https://doi.org/10.1038/s42003-020-0772-0　　**OPEN**

# Structural insights into ATP hydrolysis by the MoxR ATPase RavA and the LdcI-RavA cage-like complex

Matthew Jessop [1,3], Benoit Arragain [1,3], Roger Miras[2], Angélique Fraudeau[1], Karine Huard[1], Maria Bacia-Verloop[1], Patrice Catty[2], Jan Felix [1*], Hélène Malet [1*] & Irina Gutsche [1*]

The hexameric MoxR AAA+ ATPase RavA and the decameric lysine decarboxylase LdcI form a 3.3 MDa cage, proposed to assist assembly of specific respiratory complexes in *E. coli*. Here, we show that inside the LdcI-RavA cage, RavA hexamers adopt an asymmetric spiral conformation in which the nucleotide-free seam is constrained to two opposite orientations. Cryo-EM reconstructions of free RavA reveal two co-existing structural states: an asymmetric spiral, and a flat C2-symmetric closed ring characterised by two nucleotide-free seams. The closed ring RavA state bears close structural similarity to the pseudo two-fold symmetric crystal structure of the AAA+ unfoldase ClpX, suggesting a common ATPase mechanism. Based on these structures, and in light of the current knowledge regarding AAA+ ATPases, we propose different scenarios for the ATP hydrolysis cycle of free RavA and the LdcI-RavA cage-like complex, and extend the comparison to other AAA+ ATPases of clade 7.

[1] Institut de Biologie Structurale, Univ. Grenoble Alpes, CEA, CNRS, IBS, 71 Avenue des martyrs, F-38044 Grenoble, France. [2] Laboratoire de Chimie et Biologie des Métaux, Univ. Grenoble Alpes, CEA, CNRS, DRF, IRIG, UMR 5249, 17 rue des Martyrs, F-38054 Grenoble, France. [3] These authors contributed equally: Matthew Jessop, Benoit Arragain. *email: jan.felix@ibs.fr; helene.malet@ibs.fr; irina.gutsche@ibs.fr

AAA+ ATPases of the MoxR family are ubiquitous and found in all major phyla of bacteria and archaea. They are proposed to fulfil chaperone-like functions assisting the maturation or assembly of metabolic protein complexes[1,2], and are often found in an operon upstream of a gene encoding a von Willebrand factor A (VWA) domain-containing protein. Recent examples include the *P. denitrificans* genes *norQ* and *norD*, which code for a MoxR ATPase and a VWA domain-containing protein facilitating the insertion of the non-heme Fe$_B$ cofactor into nitric oxide reductase[3], and the *A. ferrooxidans* MoxR-related protein CbbQ which binds the VWA domain-containing CbbO to activate Ribulose-1,5-bisphosphate carboxylase/oxygenase (Rubisco)[4,5]. The most well-characterised representative of the MoxR family is the *E. coli* ATPase RavA, encoded by the *ravAviaA* operon, together with the VWA domain-containing protein ViaA. These two proteins were proposed to play a role in the maturation of both respiratory Complex I and fumarate reductase[6,7]. In addition, RavA is involved in the *E. coli* acid stress response by binding to the acid stress-inducible lysine decarboxylase LdcI[8,9]. LdcI catalyses the conversion of lysine into cadaverine, thereby consuming a proton and buffering both the intra- and extracellular medium[10,11]. Under conditions of combined acid and nutrient stress, LdcI is inhibited by the stringent response alarmone ppGpp, preventing excessive consumption of lysine[12]. However, binding of RavA to LdcI was shown to alleviate this inhibition[8]. Remarkably, RavA and LdcI together form a 3.3 megadalton cage-like complex, consisting of two D5-symmetric decameric LdcI rings located at the top and bottom of the cage, surrounded by five RavA hexamers[9].

Combined with crystal structures of the LdcI decamer (PDB ID: 3N75) and the RavA monomer (PDB ID: 3NBX), our first low resolution cryo-electron microscopy (cryo-EM) map of the LdcI–RavA cage, performed imposing the D5 symmetry of the LdcI onto the whole assembly (EMD-2679), provided initial insights into the elements involved in the complex formation[13]. Specifically, rotations of the C-terminal arms of RavA with respect to the N-terminal AAA+ ATPase modules, and accompanying massive reorientation of the tip domains called LARA (LdcI Associating domain of RavA), were shown to mediate RavA binding to either LdcI or adjacent RavA monomers in the cage[13]. The lateral contacts observed between neighbouring RavA hexamers in the LdcI–RavA complex are unique amongst AAA+ ATPases.

Building further upon these results, we now present a higher resolution cryo-EM structure of the LdcI–RavA cage in the presence of ADP, obtained without symmetry imposition. We show that the complex is built by five RavA hexamers arranged into spirals, with a prominent gap (or "seam") between two LdcI-binding RavA monomers facing either the top or the bottom LdcI decamer. Spiral conformations have recently been observed for AAA+ ATPases such as katanin, Vps4, Hsp104, ClpB and Lon[14–17], but have not yet been described for the MoxR family. In addition, cryo-EM analysis of free RavA in the presence of ADP reveals the presence of two distinct conformational states: a RavA spiral containing a single seam, equivalent to the one inside the LdcI–RavA cage, and a planar C2-symmetric ring with two nucleotide-free seams at opposite positions in the RavA hexamer. This second conformation may represent an intermediate state between the "seam up" and "seam down"-oriented RavA spirals inside the LdcI–RavA complex. Moreover, it displays remarkable structural similarity to the approximately two-fold symmetric "dimer of trimers" arrangement of subunits in crystal structures of the extensively studied AAA+ unfoldase ClpX[18,19] and the protein-remodeling AAA+ ATPase PCH2[20]. Consequently, the mechanism of the RavA ATPase cycle may be unexpectedly similar to the meticulously dissected ATP hydrolysis cycle

of ClpX[19,21–23], although the respective families of these two proteins belong to different clades of AAA+ ATPases[24–26]. Finally, we characterise the LdcI–RavA interaction using bio-layer interferometry (BLI) binding studies and ATPase activity assays. We demonstrate that while the affinity of LdcI for RavA is pH-independent, LdcI-binding results in an increase in RavA ATPase activity at acidic pH, at which this complex should be formed inside the cell. Based on these results, we propose different possible scenarios for the ATP hydrolysis cycle of RavA, both alone and in the context of the LdcI–RavA cage, and discuss their functional implications.

## Results

**The LdcI–RavA cage is formed by spiral RavA hexamers.** Initial attempts to reconstruct the LdcI–RavA complex by imposing C5 symmetry resulted in maps with visible heterogeneity for the five RavA copies (Supplementary Fig. 1). Therefore, we applied a symmetry expansion procedure (Supplementary Fig. 1, Methods), followed by a masked 3D classification without angular search using a soft mask focussing on one RavA hexamer and two LdcI decamers. This resulted in two essentially identical classes, apart from a 180° rotation around the centre of the RavA hexamer. These classes displayed left-handed spiral RavA hexamers containing a seam pointing either to the top (orientation A) or bottom (orientation B) LdcI decamer in the cage (Fig. 1, Supplementary Fig. 1). The particles from orientation A and B were grouped together after applying a 180° rotation to orientation B, and used in a second masked 3D refinement. To account for observed heterogeneity in the LARA domains of RavA, a final round of 3D classification was carried out followed by 3D refinement. The resulting two classes, with an overall resolution of 7.6 Å and 7.8 Å, respectively (Fig. 1, Supplementary Fig. 2, Supplementary Table 1), both display spiral RavA hexamers bound to LdcI, but show clear differences in the presence (Class 1) or absence (Class 2) of density for one specific LARA domain (Fig. 1c and d, panels f and h). The distance between the two LdcI decamers is about 10 Å larger in Class 2, which displays a 6° tilt between opposite LdcI rings in contrast to the parallel position of LdcI copies in Class 1. Both classes show a higher overall resolution for LdcI (5–6 Å), compared to RavA (12 Å), likely originating from the inherent flexibility of the RavA spirals in the cage (Supplementary Fig. 2). The complete LdcI–RavA cage is formed by two parallel LdcI decamers surrounded by five hexameric RavA spirals. Therefore, to illustrate the overall architecture of the LdcI–RavA cage, we constructed a C5-symmetrised map of Class 1 (Methods). Each RavA hexamer harbours six binding interfaces: two lateral interactions with neighbouring RavA monomers mediated by the triple helical domain of RavA (Fig. 1b,), and two interactions per LdcI decamer mediated by the LARA domains at the end of the four other RavA monomers (Fig. 1). The resulting map differs dramatically from the previously published one[13] (EMD-2679, PDB ID: 4UPB) that was calculated with a D5 symmetry inherent to LdcI, thereby leading to a distortion of the RavA spirals into a C2-symmetrical assembly.

A pseudo-atomic model of the cage was then created by flexibly fitting crystal structures of LdcI and RavA into the maps of Class 1 and 2 using iMODFIT[27] (Methods). The positions of the LARA domains of RavA contacting LdcI were inferred from a cryo-EM map of the LdcI-LARA complex (EMD-3206)[28]. In contrast to what was anticipated from the LdcI-LARA cryo-EM map, the crystal structure of the LdcI decamers (PDB ID: 3N75)[12] remained virtually unchanged upon fitting into the map of the LdcI–RavA complex, indicating that RavA binding does not affect the LdcI conformation. As for the atomic model of RavA to be used for flexible fitting into the spiral RavA density inside the

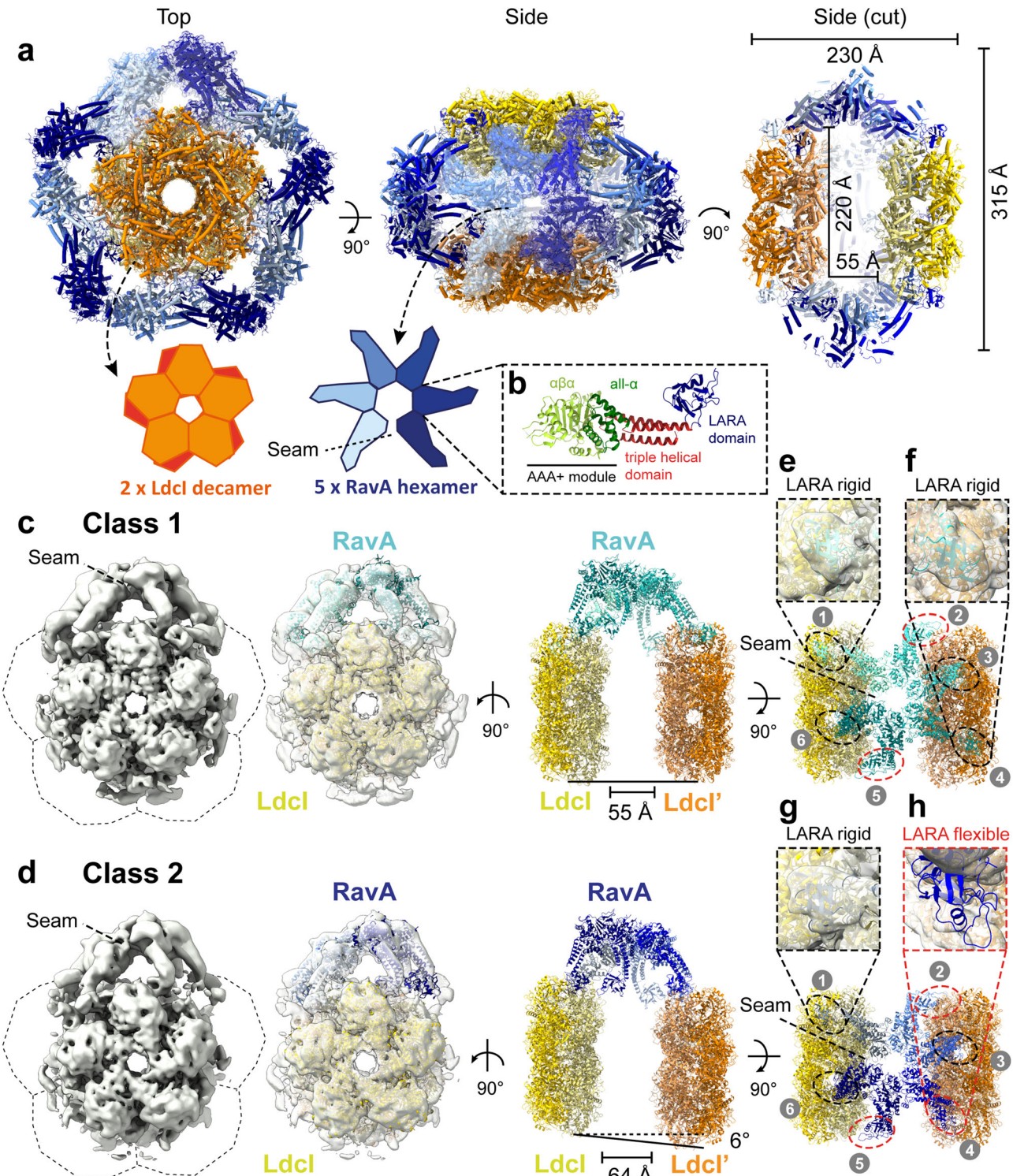

**Fig. 1 Cryo-EM structure of the LdcI–RavA cage-like complex. a** Pseudo-atomic model of the LdcI–RavA complex, based on flexible fitting of crystal structures of RavA (PDB ID: 3NBX)[8] and LdcI (PDB ID: 3N75)[12]. The cryo-EM map used for fitting corresponds to the "Class 1" map after 3D classification (containing two LdcI decamers and one RavA hexamer, see panel **b**) to which C5 symmetry has been applied. Two LdcI decamers (coloured yellow and orange) and five spiral RavA hexamers (individually coloured light to dark blue) are shown as cartoons. Top and side views are shown, as well as a cut-away side view displaying the inner cavity of the cage. A dashed box (**b**) shows one RavA monomer with annotations for the different domains: AAA+ module (green), triple helical domain (red), and LARA domain (blue). The seam in the spiral RavA hexamer is indicated by a dashed line. **c**, **d** Classes 1 (**c**) and 2 (**d**) obtained after C5 symmetry expansion followed by a masked 3D classification without angular search in RELION-2.0, resulting in C1 asymmetric maps. For each class, a post-processed cryo-EM map is shown (left) along with a fit of two LdcI decamers (yellow and orange) and one RavA hexamer (Class 1: cyan, Class 2: dark blue). Dashed lines indicate the positions of the masked-out RavA hexamers during symmetry expansion. On the right, side and top views of the fits are shown, with panels (**e**)–(**h**) (dashed boxes) focusing on specific LARA domains (numbered 1–6, black circles: rigid, red circles: flexible) contacting LdcI and their corresponding fits in the EM map. Class 2 displays a 6° tilted orientation of the second LdcI decamer (coloured orange) compared to Class 1.

cage, we decided to reexamine the crystal packing in the RavA structure (PDB ID: 3NBX)[8]. Indeed, while RavA was reported to crystallise as a monomer, analysis of the crystal packing in space group P6$_5$ reveals a continuous left-handed RavA helix (Fig. 2a) with ADP molecules bound at the intersubunit interface (Supplementary Fig. 3). This interface is essentially equivalent to the one that we originally inferred from a fit of the RavA monomer crystal structure into a low resolution C6-symmetric negative stain EM map constrained by a comparison with other AAA+ ATPases[8]. Specifically, ADP is coordinated by the Walker-A and -B residues A51, K52, S53 and D114 from one RavA monomer and by Sensor 2 and R-finger residues (RavA R251 and R170, respectively) from another RavA monomer[8] (Supplementary Fig. 3). These observations favour the idea that the crystallographic RavA–RavA interface constitutes (or at least closely resembles) the biological interface. Therefore, we opted for the usage of a spiral RavA hexamer generated from the RavA crystal structure as a starting point for flexible fitting in the cryo-EM maps. Thus, despite the overall low local resolution of RavA in the LdcI–RavA cryo-EM map, the combined use of crystal structures of LdcI decamers, RavA spiral hexamers and a cryo-EM map of LdcI-LARA allows us to confidently model the LdcI–RavA complex.

A comparison between a RavA hexamer generated from the crystal structure and RavA fitted in the cryo-EM map of Class 1 is shown in Fig. 2b. While the RavA–RavA interface is retained after fitting, major differences are observed in the pitch of the RavA spiral, and in the positions of the four LARA domains contacting LdcI. Indeed, these LARA domains (numbered 1, 3, 4 and 6 in Fig. 2b, c) undergo massive rotations when compared to the crystal structure, whereas the LARA domains of the RavA monomers involved in lateral RavA interactions only show minor movements. The apparent flexibility of RavA in our cryo-EM maps is accentuated by the disappearance of density for LARA domain 4 in Class 2, while it is clearly present in Class 1 (Fig. 1c, d, panels e–h).

**RavA seams are oriented up or down in the LdcI–RavA cage.** In the LdcI–RavA complex, the RavA spiral seam is oriented between the two RavA monomers that have LARA domains interacting with LdcI. This results in a seam that always faces either towards the upper (orientation A: "seam up") or lower (orientation B: "seam down") LdcI decamer, and never towards adjacent RavA hexamers (Fig. 1, Supplementary Fig. 1). Generally, for AAA+ ATPases forming spiral assemblies, the progressive movement of the seam around the hexameric ring is shown to be functionally important[14,15,29–32]. Therefore, the occurrence of only two opposite seam orientations of RavA spirals inside the LdcI–RavA cage may be explained as follows: (i) the lateral RavA–RavA interactions impose local geometrical constraints causing ATP hydrolysis to occur solely at the active sites formed between RavA monomers 3–4, and 1–6 in the hexamer (Fig. 2c), or (ii) binding to LdcI stalls RavA in an inactive form by preventing ATP hydrolysis from proceeding

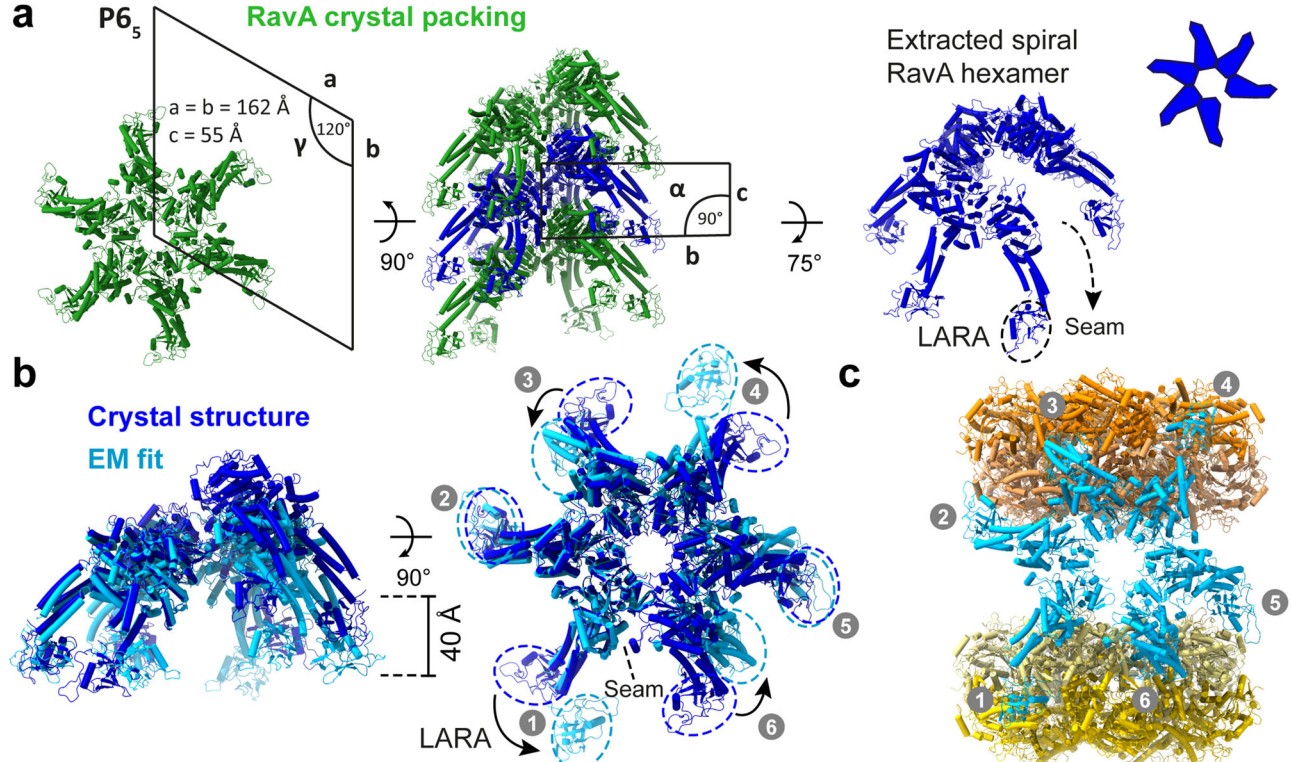

**Fig. 2 Comparison between a RavA hexamer generated from the RavA crystal structure and a fit in the cryo-EM map of the LdcI–RavA complex.**
**a** Crystal structure of *E. coli* RavA (PDB ID: 3NBX)[12], displayed as cartoons, showing the helical crystal packing of RavA crystallised in spacegroup P6$_5$. A top view along the helical screw axis of the assembly (left, with annotated unit-cell parameters) resembles a RavA hexamer. A side view of the helical assembly is shown as well (middle, with annotated unit-cell parameters), with one spiral RavA hexamer coloured dark blue. On the right, an extracted spiral hexamer (dark blue) is displayed with a slight tilt to allow visualisation of the seam, along with a schematic representation. **b** Comparison of a spiral RavA hexamer extracted from the crystal structure (dark blue) and a fit of RavA in the cryo-EM map of Class 1 (light blue). Side (left) and top views (right) are shown, with dashed circles around the LARA domains (numbered 1–6) to highlight the differences between the crystal structure and EM fit. The position of the seam is indicated by a dashed line. **c** LdcI–RavA complex obtained after fitting of structures of RavA (light blue) and LdcI (yellow and orange) in the cryo-EM map of Class 1, with LARA domains numbered as in (**b**).

in a progressive fashion. The former hypothesis would imply a switching between the two observed orientations of the RavA spiral seam and would therefore contradict a strictly sequential ATPase cycle for RavA. In addition, there remains a possibility that other seam orientations of RavA spirals in the LdcI–RavA cage also exist but are less stable and thus not resolved in our analysis because of the limited number of particles used. In light of the observed spiral conformation of RavA inside the LdcI–RavA complex, we then revisited the structure of the free RavA hexamer. Indeed, the previous negative stain EM reconstruction of the hexameric RavA was performed with a C6 symmetry imposed[8], based on planar symmetric hexamers observed in numerous AAA+ ATPase crystal structures available at the time[2,8,33–35].

**Free RavA has a spiral and a flat C2-symmetric conformation.**
Initial processing of the RavA-ADP dataset indicated a strongly preferred top-view orientation of the particles on the cryo-EM grid, resulting in a nonuniform distribution of angular projections. A second dataset was therefore collected at a 30° tilt (see "Methods"). 2D class averages from the combined dataset revealed the presence of two distinct conformational states of the RavA hexamer, containing either one (Fig. 3a, b, blue squares) or two gaps (Fig. 3a, b, red squares). 3D classification and refinement (see "Methods", Supplementary Figs. 4 and 5, Supplementary Table 1) resulted in two maps (at overall resolution of 7 and 6 Å, respectively), corresponding to an open spiral with a single nucleotide-free seam (Fig. 3c, Supplementary Figs. 4 and 5) and a planar C2-symmetric closed ring characterised by two nucleotide-free seams at opposite positions (Fig. 3d, Supplementary Figs. 4 and 5). These medium resolution maps are sufficient to enable the high-confidence flexible fitting of a RavA hexamer (for the spiral) or a trimer (for the C2-symmetric ring) extracted from the RavA helix generated from the crystal structure (Fig. 4a, b, Supplementary Fig. 6). Noteworthy, while caution in interpreting fine details at the level of the interfaces and loop regions should still be exercised, rigid-body fitting of RavA monomers into the two maps without any prior knowledge leads to an interface virtually identical to the one observed in the RavA crystal structure, further validating the RavA intersubunit interface.

The spiral structure of free RavA is equivalent to its conformation in the LdcI–RavA complex (Supplementary Fig. 6a), except for the lack of a defined density for the LARA domains that are flexible in solution before binding to LdcI. The map contains densities attributed to five ADP molecules bound to the interface between each contacting RavA monomer (Fig. 4a, c–e). In contrast, the C2-symmetric conformation contains only four ADP molecules bound in the active sites between subunits 1–2, 2–3, 4–5 and 5–6 (Fig. 4b, f). The 27 Å gap between subunits 1 and 6 in the spiral conformation is much wider than the seams in the closed-ring conformation, meaning that rearrangements between subunits 1–6 and 3–4 that destroy the nucleotide-binding site are more subtle (Supplementary Fig. 6b). Loss of nucleotide binding mainly results from a rigid-body rotation of RavA monomers 1 and 4, accounted by shifts of helix α3 and its preceding loop, which contains the Walker-A residues A51, K52 and S53, and helix α7, which contains residue M189. All of these residues directly interact with ADP in the intact active site interface (Supplementary Fig. 3, Supplementary Fig. 6c, d).

In an attempt to mimic the active ATP-bound state of the RavA hexamer, a cryo-EM dataset was collected on free RavA in the presence of ATPγS, a slowly-hydrolysable ATP analogue often used to stabilise the ATP-bound state of ATPases. However, 2D class averages indicated that RavA-ATPγS displays even more conformational heterogeneity than RavA-ADP (Supplementary

Fig. 7). Both the C2-symmetric and spiral conformations are present, as well as a seemingly C6-symmetric ring and even a C7-symmetric oligomer. The biological relevance of these two additional states is uncertain (see for instance Sysoeva, 2017[36] for review), and this heterogeneity, coupled with a strongly preferential orientation, hampers successful 3D separation. However, a comparison of 2D class averages displaying the asymmetric spiral and C2-symmetric closed ring conformations of RavA-ADP and RavA-ATPγS datasets does not reveal any noticeable differences. Most importantly, this observation suggests that, similarly to ADP, ATPγS binding does not fix RavA in one particular structural state.

**Structural insights into the ATPase cycle of RavA.** The crystal structure of the RavA monomer[8] corroborated the former phylogeny-based classification of the MoxR family as a member of the AAA+ clade 7[24–26,37]. This clade harbours in particular an additional linker (termed the pre-Sensor 2 insertion) that repositions the C-terminal helical lid of the AAA+ module relative to the N-terminal αβα core domain[25,26]. In such a spatial configuration, different from all other AAA+ clades and unique to clade 7, the Sensor 2 motif cannot contribute to ATP binding and hydrolysis in the same monomer (Supplementary Fig. 8). However, based on the crystal structure of the first crystallised clade 7 member, the magnesium chelatase BchI monomer (PDB ID: 1G8P)[38], aligned onto active hexamers from other clades, AAA+ ATPases of clade 7 were proposed to rely on a trans-acting Sensor 2 contributed by the neighbouring monomer in the hexamer[25]. The first pseudo-atomic model of the RavA hexamer, which was based on a fit of the monomer structure into a negative stain C6-symmetric EM map[2,8] and guided notably by relative positioning of Sensor 2, agreed with this hypothesis and suggested that oligomerisation of MoxR ATPases is required for completion of their ATP binding sites. Specifically, in MoxR-type AAA+ ATPases, the ATP binding site was proposed to be located not between the large (αβα) and small (all-α) AAA+ domains of the same monomer, but between the large domain of one monomer in a hexamer and the small domain of its neighbour[2,8]. The cryo-EM maps presented here and the resulting atomic models of the RavA hexamer in spiral and C2-symmetric conformations provide strong experimental support to this model, which is presently extended to all clade 7 members[25,26].

Viewed from this perspective, the planar double-seam conformation of the RavA hexamer is strikingly reminiscent of the approximately two-fold symmetric "dimer of trimers" arrangement of subunits in hexamers of the AAA+ unfoldase ClpX[18,19], which belongs to clade 5 AAA+ ATPases and thus lacks the pre-Sensor 2 insertion[25,26]. In crystal structures of ClpX, hexamers are arranged with an approximate two-fold symmetry, and contain four ClpX subunits in a nucleotide loadable (L) and two in unloadable (U) conformation on opposite sides of the hexamer. In the unloadable ClpX subunits, the small and large AAA+ domains are positioned in an "open" conformation which destroys the nucleotide-binding site[18,19]. The resulting 4L-2U arrangement of ClpX contains a characteristic seam which runs along the hexamer centre. A comparison of the C2-symmetric closed ring conformation of RavA with the 4L-2U ClpX crystal structure reveals highly similar assemblies (Fig. 5a, b). While ClpX binds nucleotides in the interface formed between the large and small AAA+ domains within one subunit, the nucleotide-binding interface in RavA hexamers is formed in between adjacent monomers (Fig. 5c, d). Importantly, our structural comparison shows that similar rigid-body like movements between the large and small AAA+ subdomains in a single ClpX subunit, or between the large or small subdomains of adjacent RavA monomers, lie at the basis of

**Fig. 3 Cryo-EM analysis of free RavA in the presence of nucleotide (ADP).** **a**, **b** 2D classes of untilted (**a**) and 30° tilted (**b**) datasets of free RavA in the presence of ADP. Red squares highlight classes belonging to a spiral RavA conformation, while blue squares show classes belonging to a C2-symmetric closed ring conformation of RavA. **c**, **d** 3D reconstructions of the spiral (**c**) and closed ring (**d**) RavA conformations corresponding to class 1 and 2, respectively. Individual subunits in the maps are coloured according to a rainbow colour scheme. The nucleotide-free seams in the two maps are annotated using dotted arrows.

L to U subunit conversion, resulting in an impaired nucleotide-binding site (Fig. 5e, f). In addition, similar C2-symmetric closed ring hexamer conformations have been observed for the crystal structures of the *T. maritima* metalloprotease FtsH[39], and the *C. elegans* protein-remodeling AAA+ ATPase PCH2, a TRIP13 ortholog[20]. Interestingly, a recent structure of human TRIP13 solved by cryo-EM in both apo- and substrate-bound states[40] displays a right-handed spiral, but no closed-ring conformation.

Previous studies have shown that conversion between L- and U-states is necessary to couple ATP hydrolysis to ClpX functioning, and provide evidence for a probabilistic model for L to U subunit switching upon ATP hydrolysis. In the proposed model, a 4L:2U ClpX hexamer converts to a 5L:1U hexamer in the presence of nucleotide, followed by subunit switching between L and U states in a non-sequential manner[19]. The cryo-EM analysis of RavA presented here reveals the presence of a mixture of both a spiral (5L:1U) and C2-symmetric closed ring conformation

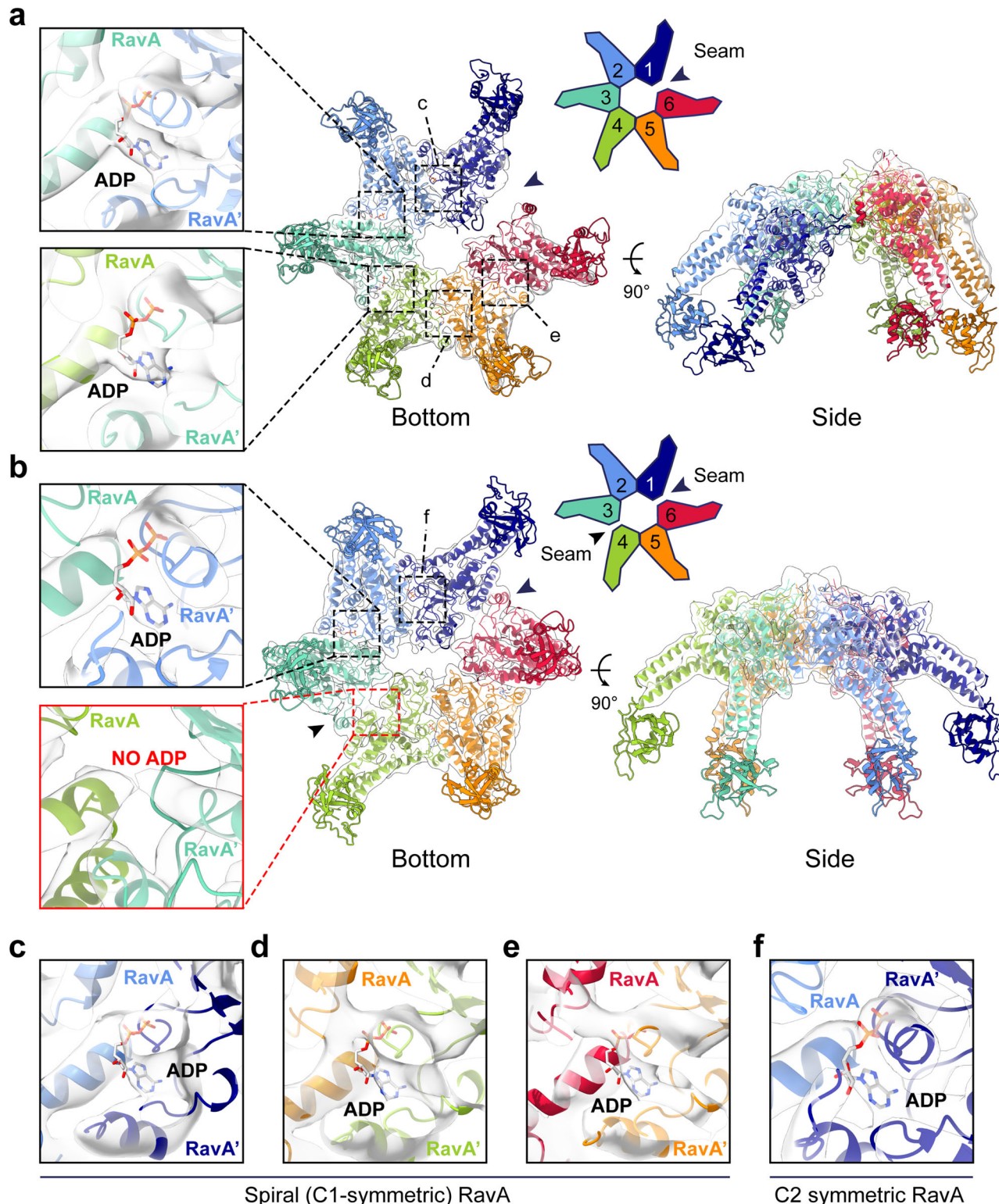

**Fig. 4 Structural analysis of spiral and C2-symmetric closed ring RavA conformations.** Fit of the spiral (**a**) and C2-symmetric closed ring (**b**) conformations of RavA in their respective EM maps, displayed as cartoons. Individual RavA subunits, labelled 1–6 in the accompanying schematic representations, are coloured according to a rainbow colour scheme. Zooms show the presence or absence of ADP in the nucleotide-binding site interface formed between subunits 2–3 and 3–4 in the spiral (**a**) and C2-symmetric closed ring (**b**) conformations of RavA. The nucleotide-free seams in the two maps are annotated using black arrows. **c–f**. Insets showing the nucleotide-binding site interface formed between subunits 1–2 (**c**), 4–5 (**d**) and 5–6 (**e**) of the spiral RavA conformation, and between subunits 1–2 (**f**) of the C2-symmetric closed ring RavA conformation.

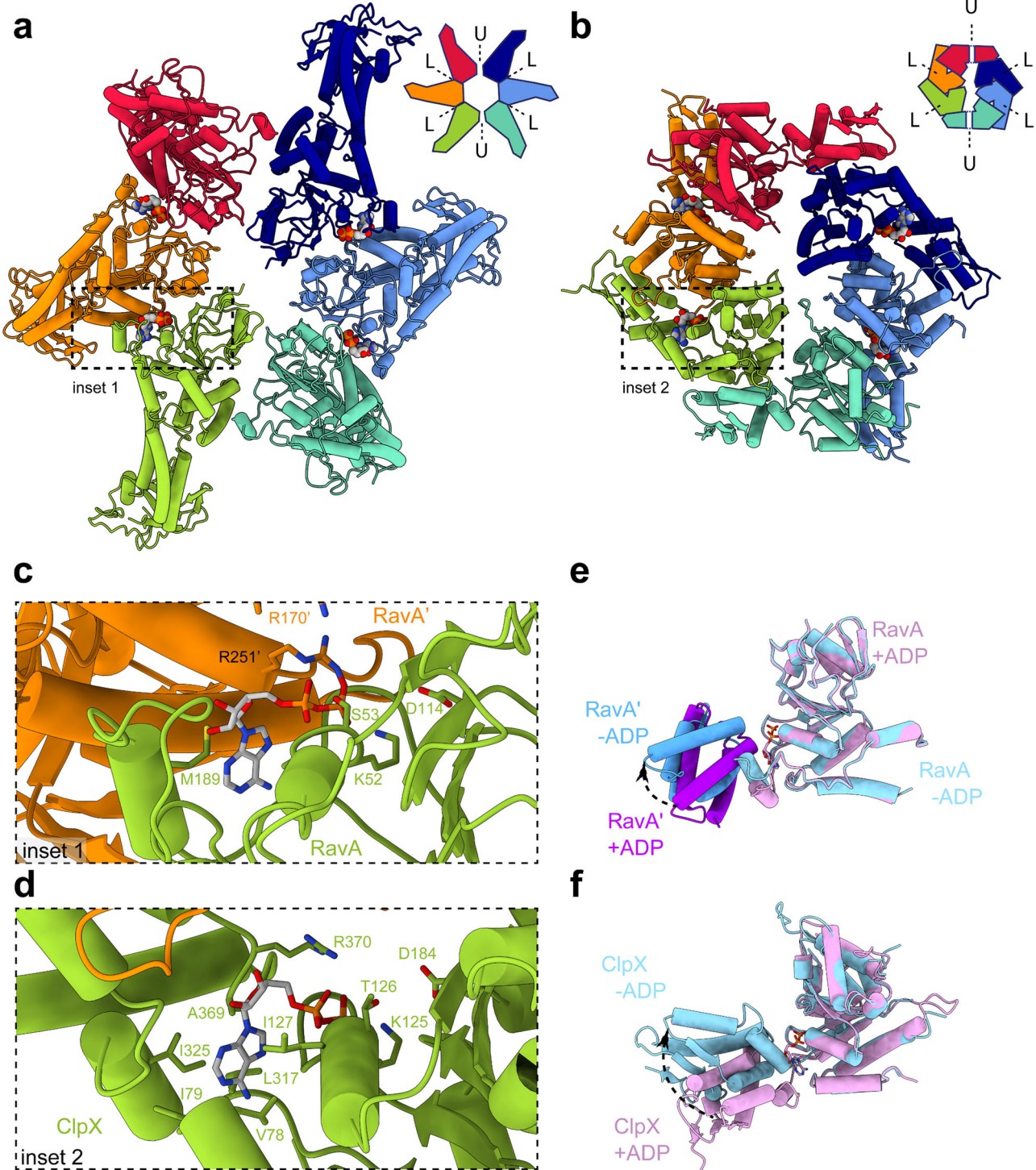

**Fig. 5 Comparison between C2-symmetric closed ring conformations of RavA and ClpX.** Comparison between closed ring conformations of RavA (**a**) and ClpX (**b**), shown as cartoons with accompanying schematic representations. RavA and ClpX subunits in equivalent positions around the hexamer are given identical colours following a rainbow colour scheme. In a RavA hexamer, the active site is formed in between the large and small AAA+ domains of adjacent RavA monomers, while in ClpX the nucleotide-binding site is formed between the large and small AAA+ domains of within a single ClpX subunit. Loadable and unloadable ATP binding sites in RavA and ClpX are annotated with L and U, respectively. **c**, **d** Zooms of the nucleotide-binding interface between adjacent RavA monomers (**c**, coloured orange and green) and the nucleotide-binding interface within one ClpX subunit (**d**, coloured green). Bound ADP molecules and interacting residues of RavA or ClpX are labelled and shown as sticks. **e** Superposition of the large and small AAA+ domains of adjacent RavA subunits (labelled RavA and RavA) from an interface in the C2-symmetric closed ring conformation with bound ADP ("closed"conformation, coloured pink and purple) and without bound ADP ("open" conformation, coloured light and dark blue). **f** Superposition of the large and small AAA+ domains within one ClpX subunit containing bound ADP ("closed" conformation, coloured pink) or without bound ADP ("open" conformation, coloured light blue). Movement of the small AAA+ domains of RavA and ClpX upon nucleotide binding is shown using black arrows.

(4L:2U) in solution. The presence of 4L:2U and 5L:1U RavA states, and the similarity between the 4L:2U conformations of ClpX and RavA, seems to suggest that RavA would function via a similar ATP hydrolysis cycle.

**LdcI binding increases RavA ATPase activity at low pH.** To investigate whether or not the restricted orientations of RavA in the LdcI–RavA complex would result in RavA being incapable of hydrolysing ATP, we performed ATPase activity measurements of RavA in the absence or presence of a three-fold molar excess of LdcI at pH values between 5.0 and 9.0. While the RavA ATPase activity increases with pH in the explored interval, LdcI increases it further at pH below 7.0, and decreases it slightly at pH 7.5 or higher (Fig. 6a, b). At pH 7.0, LdcI exerts no apparent effect on RavA ATPase activity. The observed bimodal effect of LdcI on RavA is unexpected, and contrasts with previous results which showed that LdcI increases the ATPase activity of RavA at pH 7.5[9].

Given the pH-dependent effect of LdcI on RavA ATPase activity, we investigated the LdcI–RavA interaction at different pH values using BLI. In vitro biotinylated RavA-AviTag was immobilised on streptavidin-coated biosensors, and subsequently exposed to a concentration series of LdcI at pH between 5.0 and 8.0 (see "Methods"). BLI measurements at different pH values did not show a marked difference in the binding affinity of LdcI for RavA, which ranged from $K_D$=20 nM to $K_D$=40 nM. However, we observed an increase in the height of the BLI signal (response in nm) with decreasing pH (Fig. 6c). This indicates that the mass of the bound ligand increases with lower pH, which can be explained by the pH-dependent oligomerisation of LdcI. Indeed, while LdcI is predominantly dimeric at pH 8.0, at pH 6.5 and 5.0 LdcI forms decamers and stacks of decamers, respectively (Fig. 6d)[12]. Taken together, our BLI and ATPase activity studies suggest that dimeric LdcI has a moderate inhibitory effect on RavA, while LdcI decamers and stacks increase RavA ATPase activity. Most importantly, when bound to LdcI, RavA can exert its ATPase activity over a broad pH range. Thus, despite the restricted orientations of the RavA spiral seam, RavA is still able to efficiently hydrolyse ATP in the LdcI–RavA cage.

## Discussion

AAA+ ATPases of the MoxR family have been suggested to play a role as chaperones in the assembly of multi-protein complexes, but in general the functions of MoxR family members are not well characterised and their structures are scarce[2]. Functionally, RavA has been implicated in the assembly of *E. coli* respiratory Complex I and modulation of the activity of fumarate reductase[6,7]. The crystal structure of *E. coli* RavA displays monomers that are packed in a left-handed helix, and contains an ATP binding site at the interface between adjacent monomers in the helix[8]. In fact, several other AAA+ ATPases crystallise as apparent monomers in spacegroup P6₅, thereby forming continuous helices due to crystal packing that resembles hexamers when viewed along the helical screw axis[8,29,30,32,38,41–45]. In some of these cases, the interface between monomers in the helical crystal packing is very similar to the interface elucidated by other structural methods. For instance, the spiral RavA hexamers observed in our cryo-EM reconstructions of the free RavA and the LdcI–RavA complex display an interface which is equivalent to the helical RavA assembly observed in the crystal structure. Likewise, the crystal structure of Spastin forms a helical assembly in which the monomer-monomer interface is compatible with a hexameric model based on docking of monomers in an *ab-initio* small-angle X-ray scattering envelope[42,45]. In contrast, crystal structures of Vps4 show an interface only partially similar to the interface

observed in cryo-EM maps of hexameric Vps4 spirals[30,31,46]. Moreover, the crystal structure of apo-katanin forms a helix with a different handedness than the asymmetric hexamer found in a cryo-EM reconstruction of ATP-bound katanin, and as such does not retain the biologically relevant monomer-monomer interface[29]. In addition, these helices contrast with the planar, symmetric hexamers observed in numerous available AAA+ ATPase crystal structures[5,33–35,47]. Interestingly, a crystal structure of ClpX also shows helically arranged monomers[48], and the interfaces formed are only slightly shifted compared to the ones observed in hexameric ClpX structures[18].

Based on negative stain EM data and comparison with related hexameric AAA+ ATPases, RavA was initially modelled as a planar hexameric assembly[2,8]. The cryo-EM analysis of free RavA described here unexpectedly showcases a mixture of two distinct conformational states: a spiral RavA hexamer, also observed in the present cryo-EM structure of the LdcI–RavA cage, and a flat C2-symmetric RavA hexamer characterised by two nucleotide-free seams. Spiral hexameric assemblies with a seam devoid of any bound nucleotide are common among different AAA+ ATPases[14,15,46,49]. For several AAA+ ATPases that form spiral hexamers, a sequential mechanism was proposed whereby ATP hydrolysis causes the seam to move processively around the spiral hexamer via a closed ring intermediate[14,29,49–51].

Besides a sequential ATP hydrolysis cycle, two other models are put forward to explain how AAA+ ATPases couple ATP hydrolysis to mechanical force to exert their function: the AAA+ lTag is suggested to act via concerted (all-or-none) nucleotide binding and hydrolysis that occurs simultaneously in all subunits[52], while the AAA+ unfoldases HslU and ClpX are thought to hydrolyse ATP via a probabilistic mechanism where ATP hydrolysis is not strictly sequential around the hexamer[19,21,23,53]. Evidence for probabilistic L to U subunit switching in ClpX stems from assays using individually mutated, disulfide-linked[19] or crosslinked[23] subunits in covalently tethered ClpX pseudohexamers. These studies also demonstrate that ClpX hexamers with one or more L or U locked subunits are able to hydrolyse ATP, but are impaired in substrate binding and degradation. Thus, blocking of L to U switching in a single ClpX subunit uncouples ATP hydrolysis from mechanical work, supporting a probabilistic but coordinated ATP hydrolysis mechanism in which communication between ClpX subunits is obligatory. Similar studies performed on disulfide-crosslinked HslU pseudohexamers[53] show that HslU pseudohexamers with different mixtures of active and inactive subunits can unfold protein substrates and support their degradation by HslV, albeit at a lower rate than wild-type HslUV.

Remarkably, several very recent papers reopen the debate on the exact ATPase mechanism of ClpX by showing high-resolution cryo-EM structures of the ClpXP complex where hexameric ClpX in a spiral 5L:1U state is bound to the tetradecameric ClpP protease[54–56]. Therefore, further studies are required to elucidate whether ClpXP follows a probabilistic[56] or rather processive model[55] of ATPase hydrolysis. Regardless, the existence of both spiral (5L:1U) and a C2-symmetric closed ring (4L:2U) conformations of hexameric RavA, and the resemblance of the latter to 4L:2U ClpX structures, suggests that RavA may follow a similar ATPase mechanism as proposed for ClpX. Consequently, different scenarios for RavA ATPase cycling upon LdcI binding may be envisioned. The current cryo-EM reconstructions of the LdcI–RavA complex in the presence of ADP contain five "seam up" or "seam down" RavA spirals. One possible scenario is that probabilistic L to U subunit switching allows the seam to be transferred to an opposite position in the RavA hexamer. The absence of any alternative observable seam positions other than "seam up" or "seam down" could be the result of geometrical

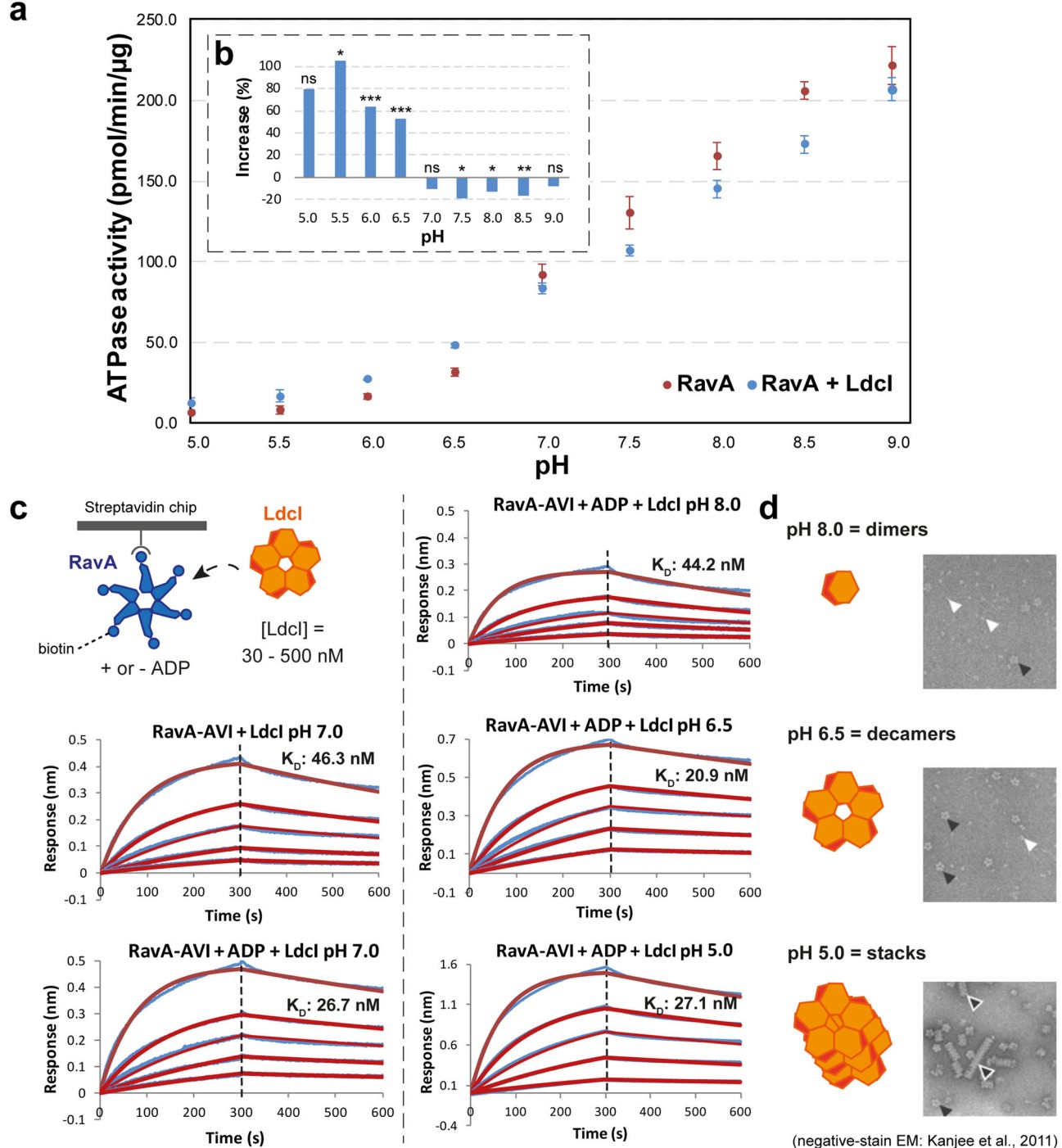

**Fig. 6 Characterisation of the LdcI–RavA interaction by ATPase assays and BLI binding studies. a** Effect of LdcI on the ATPase activity of RavA. ATPase activity was measured at various pH values (ranging from 5 to 9) as described in "Methods", either for RavA alone (red) or a mix containing RavA and a three-fold molar excess of LdcI (blue). Each data point represents the average of three independent measurements. Error bars correspond to the standard deviation. The dashed inset (**b**) shows the percent change of RavA activity when comparing the ATPase activity of RavA alone and RavA plus a three-fold molar excess of LdcI. The statistical significance is calculated using a 2-sided T-test (2-sample unequal variance). $P \geq 0.1$: not significant (ns), $P \leq 0.1$: *$P \leq 0.05$: **$P \leq 0.001$: ***. **c** BLI measurements of the LdcI–RavA interaction at different pH values ranging from 5 to 8 (blue curves: experimental data, red curves: calculated fit using a 1:1 interaction model.). For each experiment, biotinylated RavA-AVITAG was immobilised on a streptavidin-coated BLI biosensor (with or without prior incubation with 1 mM ADP) followed by binding measurements using different concentrations of LdcI (500 nM, 250 nM, 125 nM, 62.5 nM and 31.25 nM). Average values for the RavA-ADP:LdcI interaction measured at four different pH values (8, 7, 6.5, 5) are: $K_D = 29.7$ nM ± 10.0, $k_{on} = 2.68 \times 10^4$ 1/Ms ± 2.13 × $10^3$, and $k_{dis} = 8.11 \times 10^{-4}$ 1/s ± 3.41 × $10^{-4}$. The $K_D$, $k_{on}$ and $k_{dis}$ values of the individual experiments can be found in Supplementary Table 2. **d** Negative-stain EM micrographs of LdcI incubated at different pH values (reproduced from Kanjee et al.[12], with permission of the EMBO Journal)[12]. At lower pH, LdcI mainly forms decamers (pH 6.5) or stacks of decamers (pH 5), while at higher pH (pH 8) LdcI is predominantly dimeric.

constraints imposed by lateral contacts between the triple helical domains of neighbouring RavA monomers in the LdcI–RavA cage, thereby restricting the rotations between subunits needed for release of ADP by RavA monomers making these lateral contacts (schematically shown in Supplementary Fig. 9). If RavA indeed follows a similar ATP hydrolysis mechanism to ClpX, this would imply that conformational locking of RavA subunits in the LdcI–RavA cage, as described above, would lead to RavA hexamers that can hydrolyse ATP, but are functionally inactive. However, based on our data we cannot rule out the possibility that other seam orientations of RavA hexamers in the LdcI–RavA cage can occur, but are not observed in our cryo-EM analysis for example because of their transient nature. In addition, alongside the observed alternation of RavA seam states, a comparison of the cryo-EM maps representing the two classes of the LdcI–RavA complex reveals a near 10 Å difference in the distance between opposite LdcI decamers forming the central cavity of the cage (Fig. 1). If, while bound to LdcI, RavA is able to exert its potential protein-remodeling capacity, then the existence of these two classes tends to suggest that ATP hydrolysis by RavA would cause a breathing motion of the LdcI–RavA cage, thereby transferring mechanical force to remodel substrates inside the complex.

Taken together, our synergistic approach, which combines data from complementary structural techniques such as cryo-EM, X-ray crystallography, modelling, as well as biochemical characterisation, has provided insights into nucleotide-dependent conformational changes of RavA during ATP hydrolysis, and the possible ATPase mechanism of RavA in the LdcI–RavA complex. Validation of the proposed interaction partners and characterisation of interaction with substrates are necessary future steps in the elucidation of the structure-function relationships of the LdcI/RavA/ViaA triad, and in uncovering of the mechanism of its action in sensitization of *E. coli* to aminoglycosides[6]. Our work adds to the growing number of AAA+ ATPase structures corresponding to snapshots of the ATP hydrolysis cycle. Clade 7 AAA+ ATPases encompass very divergent families, including MoxR, Chelatase/YifB, the minichromosome maintenance protein MCM built by six different ATPase subunits, and even the eukaryotic Dynein/Midasin where the six AAA+ subunits are all covalently linked. It is tempting to suggest that the observations and hypotheses based on the RavA cryo-EM structures described here may be extended to all clade 7 AAA+ ATPases that share a spatial arrangement of αβα and all-α subdomain resulting in an active site formed between adjacent monomers (Supplementary Fig. 8).

## Methods

**LdcI–RavA complex formation.** RavA and LdcI proteins were expressed and purified as previously described[8,12,13], with the sole exception that LdcI was expressed in a different ppGpp$^{-/-}$ *E. coli* strain (MG1655 ΔrelA ΔspoT), generously provided by Dr. Emmanuelle Bouveret. To promote optimal LdcI–RavA cage formation in vitro, purified LdcI and RavA were initially separately diluted to a respective concentration of 0.76 mg ml$^{-1}$ and 1.2 mg ml$^{-1}$ in a buffer containing 20 mM Tris pH 7.9, 300 mM NaCl, 2 mM ADP, 10 mM MgCl$_2$, 0.1 mM PLP and 1 mM DTT. After 10 min at room temperature (RT), equal volumes of both proteins were mixed and incubated 10 min at RT. In the final mix, concentrations of LdcI and RavA were 0.38 mg ml$^{-1}$ (4.67 μM) and 0.6 mg ml$^{-1}$ (10.64 μM), respectively, resulting in a RavA monomer:LdcI monomer ratio of 10.64 μM:4.67 μM = 2.278, or approximately 4.5:2.

**Cryo-electron microscopy on the LdcI–RavA complex.** The quality of complex formation was checked by negative-stain electron microscopy (EM) using 5 times diluted 4.5:2 RavA:LdcI mix (see above). 4 μl of sample was applied to the clear side of carbon on a carbon-mica interface and stained with 1% (w/v) uranyl acetate. Images were recorded under low-dose conditions with an FEI T12 microscope operated at 120 kV or FEI F20 microscope operated at 200 kV, at nominal magnifications ranging from 13,000x to 19,000x.

For cryo-EM grid preparation, 4 μl of 4.5:2 LdcI–RavA mix was applied onto a glow-discharged quantifoil 400 mesh 1.2/1.3 grid (Quantifoil Micro Tools GmbH,

Germany), the excess solution was blotted for 3 s with a Vitrobot (FEI) using blot force 1, and the grid plunge-frozen in liquid ethane. Data collection was performed on an FEI Polara microscope operated at 300 kV. Movies of 40 frames were collected with a total exposure time of 8 s and a total dose of 40 e$^-$ Å$^{-2}$ on a K2 summit direct electron detector (Gatan) at a magnification of 41,270x, corresponding to 1.21 Å pixel$^{-1}$ at the specimen level. Specimen motion during data collection was evaluated and corrected with MotionCor2[57,58]. Frames 3–40 of each movie were dose-weighted, summed and kept for further processing. The contrast transfer function (CTF) of each micrograph was determined with GCTF[59]. 1819 best micrographs were selected based on visual quality control and CTF inspection. As previously noticed[13], and despite the high affinity of RavA for the LdcI[12], the LdcI–RavA cage is extremely sensitive to the cryo-EM grid preparation process, which results in a very low amount of intact particles per image. This difficulty in sample preparation limits the number of particles available for further analysis, complicates the particle selection process and hinders obtainment of a high-resolution structure. Eventually, 18,902 particles were manually picked using EMAN2 e2boxer[60] and subjected to two rounds of 2D classification with RELION-2.0[61] to yield a cleaned dataset containing 15,771 particles. For 2D classification and all further steps, CTF-amplitude correction was performed starting from the first peak of the CTF. Visual analysis of 2D class averages immediately revealed considerable heterogeneity in RavA conformations/positions while LdcI appeared more rigid. The initial 3D model based on 2D class averages was calculated with sxviper (SPARX)[62] imposing D5 symmetry and appeared similar to our previously published map[13]. Subsequent 3D classification in RELION-2.0 was performed with C1 symmetry in order to remove the remaining incomplete cages containing 3 or 4 RavA hexamers. This led to a clean dataset containing 11,866 particles. 3D refinement with Relion auto-refine, using the initial sxviper 3D model low-pass filtered to 40 Å as a reference and imposing C5 symmetry, led to a 7.3 Å resolution map.

Albeit already much better than our previous map[13], the resulting map showed a lower resolution of RavA in comparison to LdcI, which again pointed to structural heterogeneity of RavA particles inside the LdcI–RavA complex. Thus, a soft mask was created from a fit of one hexameric RavA[8] and two LdcI decamers[12] into the map, and the dataset was expanded by replicating each particle from the C5 consensus refinement and adding n*72° with n = 1,…,5 to its first Euler angle. A masked 3D classification was then conducted with RELION-2.0 without angular search. This procedure enabled a reliable separation of the dataset into two classes, containing 47 and 53% of the data, respectively. Unexpectedly, both classes showed left-handed RavA spirals, with a clearly defined seam facing either the upper (orientation A) or the lower (orientation B) LdcI decamer and related exactly by a 180° rotation (Supplementary Fig. 1). In order to be able to combine images of both orientations, we applied a -phi, 180°-theta, 180°+psi transformation to the Euler angles of orientation B to bring it into orientation A. Images corresponding to orientation A and rotated orientation B were then subjected to a masked local 3D auto-refinement (RELION-2.0), with the mask that again included two LdcI decamers and one RavA hexamer. This masked reconstruction had a global resolution of 7.3 Å, with local resolution ranging from 5 to 14 Å. The B-factor sharpening, using a B-factor value of −270 Å$^2$, was performed in RELION-2.0 as described[63].

To further address eventual conformational variability of the RavA spiral inside the LdcI–RavA complex, we undertook a final 3D classification using the same mask as before, including two LdcI decamers and one RavA hexamer. This classification allowed separation of two defined states (Class 1 and 2) containing 32 and 28% of particles, respectively, and a third state corresponding to more mobile/ flexible RavA containing 40% of particles (Class 3). Refinement of Class 1 and Class 2 gave respective resolutions of 7.6 and 7.8 Å, with local resolution ranging from 5 to 14 Å for both classes. In Class 1, the densities corresponding to all RavA domains contacting LdcI (i.e. LARA domains) are well resolved. In contrast, in Class 2 as well as in the preceding 7.3 Å map from the masked refinement containing all particles, the density of one LARA domain is missing.

**Post-processing of LdcI–RavA Cryo-EM maps and fitting of structures.** Local resolution estimation and subsequent filtering of maps were performed in RELION-3.0, using B-factors of −200, −250 and −300 Å$^2$ for the masked 3D refinement containing all particles, or particles from Class 1 and 2, respectively. For fitting of atomic models in the resulting filtered maps, we used the previously-determined crystal structures of LdcI (PDB ID: 3N75)[12] and RavA (PDB ID: 3NBX)[8]. Careful analysis of the RavA crystal packing revealed that RavA was crystallised as a continuous helix. In each map, two decameric LdcI molecules extracted from PDB 3N75 and one spiral RavA hexamer extracted from a continuous RavA helix generated from PDB 3NBX were first manually placed using Chimera[64], and then fitted separately using iMODFIT[27], followed by a single round of B-factor (ADP) refinement in Phenix[65,66].

**Cryo-electron microscopy on free RavA.** Purified RavA (see above) was diluted to a final concentration of 0.1 mg ml$^{-1}$ in the presence of 1 mM ADP and incubated at room temperature for 10 min. 3 μL of RavA:ADP was applied to glow-discharged (20 mA, 45 s) R2/1 400 mesh holey carbon copper grids (Quantifoil Micro Tools GmbH). Grids were plunge-frozen in liquid ethane with a Vitrobot

Mark IV (FEI) operated at 100% humidity using blot force 1 and a blot time of 2 s. Data collection was performed on an FEI Polara microscope operated at 300 kV.

A total of 2944 movies comprising 40 frames were recorded at a tilt angle of 0° on a K2 summit direct electron detector (Gatan Inc) operated in counting mode. Movies were collected with a total exposure time of 6 s and a total dose of 40 e$^-$ Å$^{-2}$. Preliminary processing suggested that RavA adopted a strongly preferred orientation on the grid. To overcome preferred orientations of RavA, a further 1083 micrographs were recorded with a 30° tilt, with a total exposure time of 6 s and a total dose of 44 e$^-$ Å$^{-2}$. All movies were recorded at a magnification of 41,270x, corresponding to a pixel size of 1.21 Å pixel$^{-1}$ at the specimen level, with a target defocus range of 1.8–3.8 μm.

RavA:ATPγS grids were prepared as for RavA:ADP, except with a 10 min incubation with ATPγS instead of ADP. Data collection was performed on a Glacios microscope (Thermo Scientific) operated at 200 kV. A total of 2809 movies (1224 of which were tilted to 30°) comprising 29 frames were recorded on a Falcon II direct electron detector (Thermo Scientific) at a magnification of 116,086×, corresponding to a pixel size of 1.206 Å pixel$^{-1}$ at the specimen level. Movies were collected with a total exposure time of 6 s and a total dose of 41 e$^-$ Å$^{-2}$, with a target defocus range of 1.5–3.5 μm.

**Image processing and 3D reconstruction of free RavA**. Motion correction on both RavA:ADP datasets was carried out using MotionCor2[57]. After discarding the first two frames, the remaining frames were aligned, dose-weighted and summed. CTF parameters were determined on aligned dose-weighted sums using CTFFIND4[67], and micrographs with an estimated resolution by CTFFIND4 of better than 8 Å were kept for further processing. Because of the observation that RavA was present as a spiral in the LdcI–RavA cage, particles were picked from all micrographs using the particle-picking software FPM[68] using a spiral hexamer extracted from the RavA crystal structure (PDB ID: 3NBX)[8] filtered to a resolution of 20 Å as a reference. Per-particle CTF estimation was then carried out on selected particles using GCTF[59] to account for variations in defocus across the tilted micrographs. A total of 924,000 particles were picked, and particles were extracted with a box size of 256 × 256 pixels. Particles from untitled and tilted micrographs were separately subjected to several rounds of 2D classification in RELION-2.0, then combined prior to 3D classification resulting in a cleaned dataset of 562,000 particles. 3D classification without imposed symmetry was subsequently carried out with four classes, using an asymmetric initial model generated in RELION-2.1 filtered to 40 Å as a reference. This resulted in three classes displaying asymmetric spirals (comprising 416,000 particles) and unexpectedly, one class showing a 2-fold symmetric closed ring (corresponding to 146,000 particles).

Particles from the three classes displaying asymmetric spirals were grouped together and subjected to a second round of 3D classification into two classes, resulting in one junk class and one good class (comprising 216,000 particles). However, density for the sixth monomer in the spiral was weak, most likely due to partial occupancy or flexibility. To resolve this monomer, a final round of 3D classification was carried out into 5 classes. Particles from the best class were then subjected to 3D refinement, resulting in a map with a final resolution of 6.94 Å after post-processing and sharpening with a B-factor of −400 Å$^2$.

Due to the fact that the micrographs were originally picked using a spiral hexamer as a reference, it was possible that side views corresponding to the 2-fold symmetric closed ring were missed during the picking process. To overcome this, micrographs were re-picked using the closed-ring map filtered to a resolution of 15 Å. 1,072,943 picked particles were subjected to per-particle CTF correction followed by several rounds of 2D classification. The resulting 721,000 particles were imported into CryoSPARC[69], and divided randomly into four subsets, each containing ~180,000 particles. For each subset, particles were subject to ab-initio 3D classification (using the Ab-initio Reconstruction algorithm) into five classes with no imposed symmetry. 257,000 particles which classified into closed-ring classes were combined, and subject to a further round of asymmetric ab-inito 3D classification into two classes, resulting in one volume with visibly less stretching in the z-direction, corresponding to 72,000 particles. Particles from this class underwent a homogeneous refinement against the resulting volume, resulting in a map with a resolution of 5.96 Å after post-processing in RELION- 2.1, which was sharpened with a B-factor of −350 Å$^2$.

For the RavA:ATPγS dataset, micrographs were motion corrected using MotionCor2, after discarding the first two frames. CTF estimation was carried out using GCTF. Particles from the best 1044 micrographs after manual screening were picked using Gautomatch (http://www.mrc-lmb.cam.ac.uk/kzhang/), using a.mrcs stack containing projections of both the C2-symmetric and spiral RavA hexamers as a reference. The resulting ~477,000 particles were subject to per-particle CTF estimation using GCTF. Particles were imported into CryoSPARC and 2D classification was then carried out. Classes showed significant heterogeneity and a strongly preferred orientation, even more so than for the RavA:ADP dataset (see Supplementary Fig. 6).

**Fitting of structures and refinement**. Local resolutions of 3D reconstructions were calculated in RELION-3.0[70]. All resolution estimates are calculated using the 0.143 gold-standard Fourier shell correlation (FSC) criterion[63]. For fitting of atomic models, a RavA hexamer (for the spiral RavA conformation) and two RavA trimers (for the C2-symmetric RavA conformation) were extracted from a continuous RavA helix generated from the crystal structure (PDB ID: 3NBX)[8], and were fitted into the corresponding maps using iMODFIT[27]. The two resulting models were then subjected to a single round of ADP refinement in Phenix, followed by geometry minimisation[65,66]. Considering the medium resolution of the cryo-EM maps for both LdcI–RavA and RavA alone, we took particular caution not to interpret the models at atomic level. Rather, we focus on large-scale conformational changes such as the orientation of the RavA seam, the distance between two LdcI decamers forming the LdcI–RavA complex, the movement of the LARA domains, the spiral and the C2-symmetric conformation of RavA and the presence or absence of the nucleotide in the RavA intersubunit interface.

**BLI binding studies**. For BLI binding studies, a C-terminal AviTag was added to RavA cloned in the p11 vector (N-terminal cleavable HIS-tag). The AviTag-containing RavA was expressed and purified using the same protocol as described for RavA[8] with the exception that 100 μM of D-biotin was added to the LB medium during expression in *E. coli* BL-21 DE3 cells (overnight expression, 20 °C). Biotinylated RavA-AviTag was purified to homogeneity, concentrated to 9 mg ml$^{-1}$, aliquoted and flash-frozen for later use. BLI experiments were performed in either 1x TBS pH 8 (25 mM Tris, 300 mM NaCl, 10 mM MgCl$_2$, 10% glycerol), 1× HBS pH 7 (25 mM HEPES, 300 mM NaCl, 10 mM MgCl$_2$, 10% glycerol), 1× MES pH 6.5 (25 mM MES, 300 mM NaCl, 10 mM MgCl$_2$, 10% glycerol) or 1× MES pH 5 supplemented with 1× kinetics buffer (0.1% w/v BSA, 0.02% v/v Tween-20), 1 mM ADP, 1 mM DTT and 0.1 mM PLP.

Experiments were performed using an Octet RED96 instrument (FortéBio), operated at 293 K. Before the start of each BLI experiment, RavA-AviTag was incubated with 1 mM ADP for 10 min. Streptavidin-coated Octet biosensors (FortéBio) were functionalised with biotinylated RavA-AviTag, quenched with 10 μg ml$^{-1}$ biocytin, and dipped in wells containing 500, 250, 125, 62.5, 31.25 or 0 nM LdcI. To check for nonspecific binding during the experiments, non-functionalised biosensors were used to measure the signal from the highest ligand concentration as well as running buffer. All data were fitted with the FortéBio Data Analysis 9.0 software using a 1:1 interaction model. Average values and standard deviations for the RavA-ADP:LdcI interaction measured at four different pH values (8, 7, 6.5, 5) are: $K_D$= 29.7 ± 10.0 nM, $k_{on}$= 2.68 × 10$^4$ ± 2.13 × 10$^3$ M$^{-1}$ s$^{-1}$, and $k_{dis}$ = 8.11 × 10$^{-4}$ ± 3.41 × 10$^{-4}$ s$^{-1}$ (Supplementary Table 2).

**RavA ATPase activity measurements**. RavA ATPase activity was measured at 25 °C on a Infinite® 200 microplate reader (TECAN). The reaction was performed in 100 μl of 50 mM MES/Tris pH 5–9, 10 mM MgCl$_2$, 2 mM ATP, 0.02% (v/v) Triton X-100, 1 mM DTT, 17 μg ml$^{-1}$ RavA. Experiments were initiated by the addition of RavA with or without a three-fold molar excess of LdcI, and stopped after 45 min by addition of 50 μl of 3% (w/v) lithium dodecyl sulfate. The amount of inorganic phosphate produced by ATP hydrolysis was assessed using the colorimetric Fiske and Subbarow method[71]. Inorganic phosphate standards were used at each of the experimental conditions to calculate the specific activity of RavA. All measurements were performed in triplicate (technical replicates).

**Statistics and reproducibility**. For BLI binding studies, the reported average value and standard deviation for the RavA:LdcI interaction were calculated using four measurements at pH 8, 7, 6.5 and 5.

For RavA ATPase activity measurements, the reported average values and standard deviations were calculated using three technical replicates per measurement.

**Reporting summary**. Further information on research design is available in the Nature Research Reporting Summary linked to this article.

## Data availability
Cryo-EM maps of the two classes of the LdcI–RavA complex in ADP-bound state, and the spiral and C2-symmetric closed ring conformations of free RavA in the presence of ADP, together with their corresponding fitted atomic structures have been submitted to the EMDB and PDB with accession codes EMD-4469 and PDB-6Q7L for LdcI–RavA Class 1, EMD-4470 and PDB-6Q7M for LdcI–RavA Class 2, EMD-10351 and PDB-6SZA for the C2-symmetric closed ring RavA conformation, and EMD-10352 and PDB-6SZB for the spiral RavA conformation. Source Data for Fig. 6a–c can be found in Supplementary Data 1.

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

## Acknowledgements

We thank Guy Schoehn for establishing and managing the cryo-electron microscopy platform and for providing training and support. We are grateful to Aymeric Peuch for help with the usage of the EM computing cluster and to Ambroise Desfosses and Leandro Estrozi for discussions. This work was funded by the European Union's Horizon 2020 research and innovation programme under grant agreement No 647784 to IG. For electron microscopy, this work used the platforms of the Grenoble Instruct-ERIC Center (ISBG: UMS 3518 CNRS-CEA-UGA-EMBL) with support from FRISBI (ANR-10-INSB-05-02) and GRAL, a project of the University Grenoble Alpes graduate school (Ecoles Universitaires de Recherche) CBH-EUR-GS (ANR-17-EURE-0003). IBS acknowledges integration into the Interdisciplinary Research Institute of Grenoble (IRIG, CEA). The IBS electron microscope facility is supported by the Rhône-Alpes Region, the Fondation pour la Recherche Médicale (FRM), the fonds FEDER, the Centre National de la Recherche Scientifique (CNRS), the Commissariat à l'Energie Atomique et aux Energies Alternatives (CEA), the University of Grenoble Alpes, EMBL, and the GIS-Infrastructures en Biologie Santé et Agronomie (IBISA). MJ was funded by a CEA IRTELIS PhD fellowship, JF was supported by a long-term EMBO fellowship (ALTF441-2017) and a Marie Skłodowska-Curie actions Individual Fellowship (789385, RespViRALI).

## Author contributions

M.J., B.A., A.F., K.H., J.F. and H.M. purified proteins. M.J., B.A., M.B.V., J.F., H.M. and I.G. made cryo-E.M. grids. M.J., B.A., M.B.V., H.M. and I.G. collected E.M. data. M.J., B.A., J.F., H.M. and I.G. analysed E.M. data. R.M. and P.C. performed ATPase activity measurements. M.J. and J.F. performed B.L.I. measurements. M.J., J.F. and I.G. analysed structures and mechanisms. H.M. supervised B.A. and contributed to the initial design of the project together with I.G. I.G. designed, directed and funded the overall study. J.F. and I.G. wrote the paper with significant input of M.J. and H.M.

## Competing interests

The authors declare no competing interests.
