## [Peer Review File · Communications Biology]

Editorial Note: *This manuscript has been previously reviewed at another Nature Research journal. This document only contains reviewer comments and rebuttal letters for versions considered at Communications Biology.*

REVIEWERS' COMMENTS:

Reviewer #1 (Remarks to the Author):

In my opinion the authors have thoroughly addressed all points raised by the reviewers. I recommend publication in "Communications Biology".

Reviewer #2 (Remarks to the Author):

Jessop et al. have addressed some of my concerns. However, my main criticism that the map quality and the resolution of the RavA maps is not state of the art remains. I think this is a serious concern, because other AAA-ATPases have been resolved at much higher resolution even if processed with a C1-symmetry. This suggests that there are technical obstacles, which the authors cannot overcome but others have overcome in their studies. Especially the spiral conformation remains distorted, due to the anisotropic resolution. This is evident by the badly resolved helices in the side view orientation. The distortion caused by the anisotropy in resolution will influence the model building and can lead to distortions in the model. The authors did nothing to overcome this shortcoming in their revised version.

I agree with the authors that low resolution maps can be used for modelling and that the resulting models can be close to the ground truth. The detected gross rearrangements are probably largely true. Details in contact sites or loop-conformations might need further experimental validation. Such models are probably better regarded as illustrations or interpretations of a map than as solid results.

REVIEWERS' COMMENTS:

Reviewer #1 (Remarks to the Author):

In my opinion the authors have thoroughly addressed all points raised by the reviewers. I recommend publication in "Communications Biology".

Reviewer #2 (Remarks to the Author):

Jessop et al. have addressed some of my concerns. However, my main criticism that the map quality and the resolution of the RavA maps is not state of the art remains. I think this is a serious concern, because other AAA-ATPases have been resolved at much higher resolution even if processed with a C1-symmetry. This suggests that there are technical obstacles, which the authors cannot overcome but others have overcome in their studies.

We are fully aware that, in principle, these technical obstacles could be overcome by the use of detergents, graphene grids or substrates for example. This would however have necessitated an important extra effort of data collection and analysis that we considered unnecessary at this point, because we previously obtained a crystal structure of RavA in which the monomers are arranged in a way preserving the intersubunit interface and the nucleotide binding site. Therefore, our 6-7 Å resolution maps are fully sufficient to derive models and draw conclusions that were the object of the present manuscript, as long as the data is not carelessly overinterpreted.

Especially the spiral conformation remains distorted, due to the anisotropic resolution. This is evident by the badly resolved helices in the side view orientation. The distortion caused by the anisotropy in resolution will influence the model building and can lead to distortions in the model. The authors did nothing to overcome this shortcoming in their revised version.

See explanation above. The spiral conformation is present in our crystal structure, and even if the cryo-EM reconstruction of the spiral conformation could have become less anisotropic if a new dataset with a uniform distributed particle orientations could have been collected, the fit of the crystal structure into the map (which retains the intersubunit interfaces as described) would have remained the same.

We understand that nowadays the access to high-end cryo-EM microscopes has become easier and that, in principle, it is always possible to collect more and better data. However, we are truly convinced that the effort should be guided not primarily by the goal of increasing the nominal resolution, but by the question of whether an increase in resolution would bring significant advances in the map interpretation. We believe that in this particular work, considering the presence of the crystal structure, investing our effort in an improvement of the nominal resolution would not have been justified. We preferred to invest this effort in a synergistic usage of cryo-EM and X-ray crystallography in order to propose a molecular model for ATP hydrolysis by clade 7 AAA+ ATPases.

I agree with the authors that low resolution maps can be used for modelling and that the resulting models can be close to the ground truth. The detected gross rearrangements are probably largely true. Details in contact sites or loop-conformations might need further experimental validation. Such models are probably better regarded as illustrations or interpretations of a map than as solid results.

To address this criticism, we added following sentence to the manuscript: « Noteworthy, while caution in interpreting fine details at the level of the interfaces and loop regions should still be

exercised, rigid body fitting of RavA monomers into the two maps without any prior knowledge leads to an interface virtually identical to the one observed in the RavA crystal structure, further validating the RavA intersubunit interface. »